# Matrix Metalloproteinases in Age-Related Macular Degeneration (AMD)

**DOI:** 10.3390/ijms21165934

**Published:** 2020-08-18

**Authors:** Luis García-Onrubia, Fco. Javier Valentín-Bravo, Rosa M. Coco-Martin, Rogelio González-Sarmiento, J. Carlos Pastor, Ricardo Usategui-Martín, Salvador Pastor-Idoate

**Affiliations:** 1Clinical University Hospital of Valladolid, Av. Ramón y Cajal, 3, 47003 Valladolid, Spain; luisonrubia91@gmail.com (L.G.-O.); franciscojavier.valentin@gmail.com (F.J.V.-B.); pastor@ioba.med.uva.es (J.C.P.); 2Institute of Applied Ophthalmobiology (IOBA), University of Valladolid, 47011 Valladolid, Spain; rosa@ioba.med.uva.es; 3Cooperative Health Network for Research in Ophthalmology (Oftared), National Institute of Health Carlos III, ISCIII, 28040 Madrid, Spain; 4Institute of Biomedical Research of Salamanca (IBSAL), 37007 Salamanca, Spain; gonzalez@usal.es; 5Institute of Molecular and Cellular Biology of Cancer (IBMCC), University of Salamanca—CSIC, 37007 Salamanca, Spain

**Keywords:** age-related macular degeneration, extracellular matrix, Bruch’s membrane, matrix metalloproteinases, tissue inhibitors of metalloproteinases, MMPs polymorphisms

## Abstract

Age-related macular degeneration (AMD) is a complex, multifactorial and progressive retinal disease affecting millions of people worldwide. In developed countries, it is the leading cause of vision loss and legal blindness among the elderly. Although the pathogenesis of AMD is still barely understood, recent studies have reported that disorders in the regulation of the extracellular matrix (ECM) play an important role in its etiopathogenesis. The dynamic metabolism of the ECM is closely regulated by matrix metalloproteinases (MMPs) and the tissue inhibitors of metalloproteinases (TIMPs). The present review focuses on the crucial processes that occur at the level of the Bruch’s membrane, with special emphasis on MMPs, TIMPs, and the polymorphisms associated with increased susceptibility to AMD development. A systematic literature search was performed, covering the years 1990–2020, using the following keywords: AMD, extracellular matrix, Bruch’s membrane, MMPs, TIMPs, and MMPs polymorphisms in AMD. In both early and advanced AMD, the pathological dynamic changes of ECM structural components are caused by the dysfunction of specific regulators and by the influence of other regulatory systems connected with both genetic and environmental factors. Better insight into the pathological role of MMP/TIMP complexes may lead to the development of new strategies for AMD treatment and prevention.

## 1. Introduction

Age-related macular degeneration (AMD) is the leading cause of irreversible central loss of vision among patients over 60 years of age in developed countries; it accounts for 8.7% of all blindness worldwide, a percentage that translates to about 30–50 million people [1]. Due to the aging populations in developed countries, an increase in the number of people affected by AMD has been forecast [2], with the number expected to reach around 288 million in 2040 [3]. The loss of visual function associated with AMD negatively impacts patients’ quality of life and can lead to the loss of independence, social isolation, and absence from work. The costs associated with AMD and other types of vision impairment have significantly increased the economic burden on patients, caregivers, and society. However, despite its prevalence and the high cost of its treatment, the available therapeutic options for delaying its progression or, better yet, approaches toward minimizing or eliminating its risk factors, are very limited in number.

Nowadays, AMD is classified into an early stage, which is characterized by drusen and pigmentary changes; an intermediate stage, characterized by the presence of large drusen, abnormalities in the retinal pigment epithelium (RPE) cells, or both; and a late stage, which can be categorized into one of two subtypes (Figure 1 and Figure 2): geographic atrophy (GA), also known as the dry form, or choroidal neovascularization (CNV), also known as the wet form [1,3,4]. GA in AMD usually advances slowly, affecting progressively the photoreceptors, the RPE cells, the Bruch’s membrane (BM), and the choriocapillaris complex [5]. In general, it takes a longer time to notice the loss of central vision in patients with GA, although sometimes GA can shift to the wet form with a sudden loss of vision in a few days. Currently, there is not any efficient treatment for dry-from AMD. Neovascular AMD is characterized by the presence of CNV in the central area (about 10–15% of all patients with advanced AMD experience this) [5], causing profound and rapid loss of central vision due to recurrent retinal exudation, subretinal hemorrhage, macular detachments and, in the final stages of the disease, disciform scars. Currently, wet-form AMD is treated with repeated injections of anti-angiogenic treatments, which are effective at improving visual acuity [1,6].

Risk factors associated with AMD appear to be wide-ranging, encompassing age, lifestyle, environmental and systemic factors, as well as genetic predisposition. Age is known to be the strongest risk factor. Regarding gender, there is no consensus in the literature; some studies reported a weak association between female gender and AMD [7], while others did not find any such association [3,8,9], although a higher prevalence of drusen and late AMD in men than in women in an Asian population has been observed [7]. Ethnicity may have an effect on AMD development; Wong et al. [3] reported that AMD is more prevalent in the European than in African population, although when compared with Asians, just early AMD prevalence was found to be higher in Europeans [3]. There are other modifiable factors that have been associated with the development of AMD, such as smoking [10,11], obesity, the intake of ω-3 fatty acids, and insufficient physical exercise [12,13,14].

Although several molecular pathways are involved, such as the accumulation of oxidative products or dysregulation in the vascular and immune system [15] among others, AMD pathophysiology is yet to be fully understood. Using this framework, recent studies have reported that disorders in the regulation of the extracellular matrix (ECM) could play an important role in its etiopathogenesis, as well as its regulator systems, which are composed mainly of the matrix metalloproteinases (MMPs) and the tissue inhibitors of metalloproteinases (TIMPs) [16,17,18,19].

With advancing age, significant changes can occur in the ECM that hinder its functions, resulting in the accumulation of waste material [20]. The formation of drusen, which is the hallmark of AMD, is thought to be due to the malfunctioning of the RPE cells and the dysregulation of the remodeling of the ECM as a result of its presence in different regions in the Bruch’s membrane (BM) [21]. MMPs and TIMPs are crucial for the regulation of the ECM [22,23], and ECM dysregulation by the modification of MMP and TIMP activity could also be associated with an increased risk of AMD [5].

AMD is characterized by the loss and reduction function of the photoreceptor and RPE cells, and is associated with pathological matrix remodeling and degradation, cell proliferation, neovascularization, and chronic inflammation. The modulation of ECM turnover by changing the RPE secretion of MMPs and TIMPs may play a central role in the normal functions and pathogenesis of the retina. The pathological degradation or accumulation of ECM structural components, which may eventually lead to AMD development, is caused by a dysregulation of specific MMP/TIMP complexes, and also by the influence of other mechanisms connected with both genetic and environmental factors [18,19]. Moreover, a growing number of studies have recently reported the association between different polymorphisms of *MMP* and *TIMP* genes and AMD [24,25,26,27,28,29,30,31,32,33,34,35,36,37,38]. Therefore, not only age, but also genetic components and environmental stress factors, contribute to the occurrence of the disease, which explains why not all elderly people have AMD [39,40].

The main aim of this article is to review the relevance and impact of MMPs and TIMPs on the development of AMD and their roles as biomarkers and/or therapeutic targets. We will illustrate the activities of MMPs and TIMPs for the integrity of the ECM, the changes in the activity of MMPs expressed by RPE cells, and the different genetic variants of *MMPs* and *TIMPs*, some of which could predispose individuals to AMD. Also, this work analyzes some studies on MMP inhibitors which are already used to control MMP activity, and subsequently recommends their application as therapeutic agents for the treatment of AMD.

## 2. Methods

A comprehensive review of the literature was performed using the MEDLINE, PubMed, Web of Science, Scopus, and Embase electronic databases, covering the years 1990–2020. Potentially relevant articles were sought using the following search terms in combination as Medical Subject Heading (MeSH) terms and text words: “Age-related macular degeneration”, “extracellular matrix”, “Bruch’s membrane”, “metalloproteinases”, “tissue metalloproteinases inhibitors” and “matrix metalloproteinases polymorphisms in age-related macular degeneration”. We also studied reviews, comments, and disquisitions on the pathology. In addition, we scanned the reference lists of the retrieved publications to identify additional relevant articles. The search was supplemented using the MedLine option “Related Articles”. No language restrictions were applied.

## 3. Extracellular Matrix in the Eye and the Metalloproteinases

In the retina, the BM is a 2–4-µm thick, acellular, five-layered ECM located between the retina and choroid [41,42]. The BM is made up of 5 layers with a central elastic layer, which is mainly composed of elastin, embraced by the inner and the outer collagenous layers with a high concentration of collagens I, II and V. Two basement membranes complete the structure: the RPE basal membrane and the choriocapillaris basal lamina, whose main components are collagen type IV, fibronectin, and laminin [41,42]. Due to its location between the metabolically active RPE and the choriocapillaris, the BM acts as the scaffold for the RPE, takes part in the regulation of the diffusion of nutrients within the choroid-RPE complex, and gives structural support against neovascularization from the choroid to the avascular outer retina through antiangiogenic molecules in the elastin layer. The involvement of the RPE cells in the stability of the BM structure by controlling the synthesis of collagen types I and IV and laminin, as much as by taking part in the complex regulation of MMPs [41,43], is well known, suggesting that the BM is the site of the primary lesion in AMD [43].

The expression of most MMPs in tissues under normal conditions is low, and it is induced when remodeling of the ECM is required. Under pathological states, their expression can be increased, which may interfere with the metabolism of the retina as a whole, causing alterations in the modulation of ECM turnover and in the retinal interphotoreceptor matrix (IPM). The synthesis of MMPs undergoes complex regulation by cytokines, interleukins, growth factors and hormones, prostaglandins, and genetic factors, among others [44].

MMPs belong to a superfamily of homologous, multidomain, zinc-dependent endopeptidases, also called matrixins, i.e., the a disintegrin and metalloproteinase domain (ADAMs) and the a disintegrin and metalloproteinase with thrombospondin motifs (ADAMTSs) [45]. In humans, MMPs are made up of a family of 23 proteins, which are homogeneous in structure, function, and localization. These proteases are responsible for proteolytic processes in the BM as a result of their ability to cleavage ECM molecules, degrading substrates such as elastin, gelatin, and collagen I, IV and V. Due to the great variety of substrates with which MMPs can interact, such as cytokines, cell surface molecules, or non-ECM molecules [46], they can take part in a wide range of processes, including proteolysis, cell adhesion, angiogenesis, wound healing, inflammation, cell proliferation, and in development processes [47]. In addition, MMPs play a central role in the direct activation of signaling molecules, such as tumor necrosis factor (TNF) and other cytokines; MMPs therefore contribute to various aspects of immunity [48]. A detailed classification of MMPs and TIMPs, based on their substrates, general biological effect, localization in the eye, and processes in which they take part, is provided in the Appendix A [16,17,32,38,41,49,50,51,52,53,54,55,56,57,58,59,60,61,62,63,64,65,66,67,68,69,70,71,72,73,74,75,76,77].

ADAMs and ADAMTSs are endopeptidases related to MMPs. Whilst ADAMs primarily act as transmembrane proteins with functions in cell adhesion and the proteolytic processing of the ectodomain of cell surface receptors and signal molecules (i.e., ADAM10, ADAM12 and ADAM15), ADAMTSs are secreted proteins with procollagen activity which process and deposit collagen into the ECM and the BM at the retina level (i.e., ADAMTS-2 and ADAMTS-3). These proteases are also responsible for numerous other biological processes in the retina in normal and pathological conditions.

ADAMs have been related to the normal development of RPE cells, showing an important role in maintaining epidermal integrity. In fact, loss of the function of ADAMs causes the disorganization of RPE cells, resulting in the interruption of the photoreceptor cell functions [78,79]. Also, they are involved in inflammatory processes by the activation of a large number of substrates, including cytokine receptors, TNF receptor, EGF receptor, adhesion molecules, and transforming growth factor and retinal neovascularization by the activation of vascular endothelial growth factor (VEGF) receptor [79]. The involvement of ADAMs by VEGF-A activation in vascular endothelial cells suggests their role in regulating the neovascularization of the retina [79]. The proteolytic ability of MMPs and ADAMs is directly regulated by TIMPs, which bind to them and inhibit their activities [16].

TIMPs and 21–28 kDa proteins are the main local regulators of MMPs activity, even though other proteins have also been associated with this, such as α-macroglobulins [80], the tissue factor pathway inhibitor [81], and the secreted form of the membrane-bound β-amyloid protein [82,83]. TIMPs have both a C-terminal domain and an N-terminal domain, with each containing three conserved disulfide bonds. The N-terminal domain folds within itself and has the capacity to inhibit MMPs [48,84]. Of note, although TIMPs are highly similar in structure, they have remarkably different expression patterns. In addition, their expression varies according to different physiological stimuli in diverse cell types [48].

There are at least four members of the TIMP family that bind with MMPs in a 1:1 ratio stoichiometry [84]. They have poor specificity, whereby each one can inhibit several MMPs, but not with the same efficacy [85]. Active MMPs regulate the remodeling of the ECM due to their degradative capacity, and stimulate the secretion of TIMPs, which have an inhibitory effect on MMP activity. These processes are thought to maintain the integrity of the BM [86]. TIMP-1 inhibits the activity of the membrane-type MMPs (MT-MMPs such as MMP-14), TIMP-2 and TIMP-3 can bind to all types of MMPs and also inhibit several ADAM and ADAMTS family members, and TIMP-4 is able to inhibit the activity of MMP-1, MMP-2, MMP3, MMP-7, and MMP-9 [84,87,88]. TIMP-3 is found in chromosome 22q12.3 and is enclosed within an intron of the gene, synapsin 3 (SYN3), while TIMP-1 and TIMP-4 are located within the introns of synapsin 1 (SYN1) and synapsin 2 (SYN2), respectively; however, MMP-2 seems not to have this feature [16].

The expression of MMP and TIMP proteins is tightly regulated to maintain adequate balance between ECM synthesis and degradation, which is essential for healthy tissues. Disorders in the regulation of the MMP/TIMP complex have been implicated in many pathological conditions including cancer, Alzheimer’s, cardiovascular or rheumatologic diseases [89,90,91], as well as eye diseases such as retinal dystrophies [92,93,94], retinal detachment [95] and proliferative vitreoretinopathy (PVR) [49,96,97,98], wound corneal healing [99,100,101], glaucoma [102,103,104,105], diabetic retinopathy (DR) [106,107,108,109,110], ocular tumors [111,112,113,114], pseudoexfoliation syndrome [115,116,117], and epiretinal membranes formation [118]. It is thought that the lack of stability in the regulation of MMPs may play a key role in the pathogenesis of AMD [5,17,50,119,120,121,122,123,124,125].

## 4. Metalloproteinase and Tissue Inhibitors of the Metalloproteinase Pathway in AMD

Several studies have shown that MMPs and TIMPs play a key role in the homeostasis and changes in the ECM in the eye. However, the exact source of MMPs in the retina area is still unknown, even though the BM and RPE could release them [86]. The presence of various MMPs, mainly MMP-1, MMP-2, MMP3, and MMP-9, and TIMPs, such as TIMP-1 and TIMP-3, in the IPM, BM and RPE, has been reported [86].

Guo et al. [86] demonstrated the presence of MMP-2 and MMP-9 in the human BM and their dissimilar distribution in the retina, with lower levels of MMP-2 in the macular region. Furthermore, Guo et al. [86] identified the increased levels of active MMP-2 in the periphery as one of the possible causes of regional differences in the conductivity of the BM [126,127,128,129,130,131,132,133,134]. Plantner et al. [135] demonstrated a significant increase in the levels of MMP-2 in the RPE-associated IPM in AMD patients compared to normal donors, suggesting that MMP-2 could play a role in the pathology of AMD. Finally, Chau et al. [119] identified a possible increase in the circulation of MMP-9 in the plasma of patients with AMD.

In general, the MMP level in the BM increases proportionally with BM thickness and age [135]. Dysregulation of MMP complex has been involved in the pathogenesis of AMD, either in early AMD [120,136] or late neovascular AMD [135,137,138,139]. The main clinical sign of the early stage of AMD is the drusen, which are composed of a great variety of molecules; indeed, over 100 molecules have been identified, such as markers of inflammation (c-reactive protein), acute-phase reactants (vitronectin), complement components (factor H), race elements (including zinc, iron and calcium), apolipoproteins B and E, MMPs, TIMPs, and others [140,141,142,143], which are as a result of the imbalance in the ECM turnover [144]. The role of MMPs in such a process is of the utmost importance, as they are the principal ECM-degrading proteinase [145]. Being the main enzyme synthesized by RPE cells [50], MMP-2 seems to play a critical role in early AMD development, due to the accumulation of deposits under the RPE and the increase in collagen IV when its activity decreases [43]. Furthermore, some authors have proposed that one of the possible reasons for the increased incidence of AMD in women with decreased levels of estrogen could be the imbalance in the activity of MMP-2 [136,144] due to the action of the estrogen receptors within the RPE [146], although this assertion remains controversial.

MMPs are regulated at certain levels in four critical stages [46]: gene expression, compartmentalization, proenzyme activation, and enzyme inactivation. MMP synthesis is mainly controlled at the level of gene transcription, during which the binding of transcription factors to specific sequences (promoter regions) of the gene takes place. It is of utmost importance to note that MMPs are released in an inactive form (pro-MMPs), so an activation process is required. Thus, increased levels of inactive MMPs do not imply an increase in their proteolytic activity.

The activation of MMP-2 (Figure 3), which takes place on the RPE cell surface, requires the presence of other MMPs, the MT1-MMP, such as MMP-14, and the presence of TIMP-2 [41,51,52]. MMP-14 is one of the most studied proteolytic enzymes, with a broad substrate, especially against the ECM components. It can degrade ECM components directly or indirectly by activating pro-MMP-2. Prior to pro-MMP-2 activation, the formation of the MT1-MMP/TIMP-2 complex (binary complex) is required, which serves as a receptor for pro-MMP-2, resulting in a ternary complex. This complex is ready for activation by an MT1-MMP free of TIMP-2, which cleaves the pro-MMP-2, releasing an activated MMP-2. Thus, the appropriate regulation of the components of the trimolecular complex and their adequate concentration could play a fundamental role in the prevention of AMD, as it has been reported that MMP-14 and TIMP-2 are essential for maintaining the levels of MMP-2 activation induced by oxidative stress in RPE cells [147,148,149,150].

Furthermore, there are some competitive reactions that could lead to the reduction of pro-MMP-2, such as the interaction with pro-MMP-9, producing the high molecular weight species (HMW1 and HMW2) or even the large macromolecular weight MMP complex [41,51,52]. Hussain et al. [52] reported increased levels of pro-MMP-9, HMW1, and HMW2, with decreased levels of active MMP-2 and MMP-9, in Bruch’s-choroid preparations of AMD patients compared with a control group.

There is no consensus on the current knowledge about the involvement of MMPs in wet AMD and CNV development, with some studies supporting a protective role of MMPs [53] and others claiming the opposite [52,151]. Although the exact molecular signals in the development of choroidal neovascularization have not yet been completely elucidated, most studies [137,138,139,152,153] concur that there could be a dysregulation in the MMP complex, as choroidal neovascularization is an invasive process.

High levels of MMPs, in particular MMP-2 and MMP-9, have been found in patients with proliferative diabetic retinopathy and new blood vessels [138,152], and the strong expression of some MMPs has been demonstrated in CNV membranes surgically removed from AMD patients [154]. Steen et al. [137] reported that increased mRNA levels of wet AMD contained high levels of MMP-9 [44], and increased plasma levels of MMP-2 and MMP-9 were reported in patients with wet AMD [119], suggesting that these enzymes might contribute to the progression of choroidal angiogenesis. Because MMP-2 and MMP-9 levels increase with aging and degrade collagen type IV, many studies have focused mainly on evaluating MMP-2 and MMP-9 expression in AMD patients [137]. Nevertheless, some studies have also established that other MMPs, such as MMP-7, TIMP-3, MMP-14, and TIMP-2, might also contribute to the modulation of MMP activities in RPE cells in AMD [124,155].

Some MMPs, such as MMP-12, MMP-2, and MMP-9, have the capacity to degrade elastin. Thus, some researchers have suggested that the disruption of the elastic layer of the BM could be implicated in the development of CNV in the macular region, where it is known to be more discontinuous [42].

Sivaprasad et al. [156] studied the hypothesis that the circulation levels of soluble elastin-derived peptides (S-EDPs), which are released as a result of partial elastin proteolysis, could be higher in patients with early AMD and with wet AMD. Not only did they report higher serum levels of S-EDPs in patients with early AMD than in control subjects, but they also found increased levels of S-EDPs in patients with wet AMD compared to those with early disease. They suggested that the increased systemic elastin degradation may increase the risk of conversion from early AMD to neovascular AMD.

Furthermore, Lambert et al. [138] studied the expression and activity of MMP members in mice with laser-induced choroidal neovascularization. They found that mice with low MMP-2 and MMP-9 gene expression had lower rates of incidence, and lower severity for the relatively few incidences, of laser-induced choroidal reaction. However, the lack of concordance between the events leading to the development of CNV in human AMD patients and the events leading to the same outcome in laser-induced animal models [157], as well as the increased activity of MMPs described in wound healing and inflammation, could have biased the outcome.

The main angiogenic factor in the retina which is able to promote neovascularization is VEGF [158,159]. A feedback loop appears to exist between VEGF and MMP molecules, as changes in the proteins of the extracellular matrix can increase the secretion of VEGF by RPE cells [139,154,160], and VEGF-A, by itself, is able to upregulate the expression of MMPs in RPE cells [160]. However, how MMPs modulate the secretion of VEGF by RPE cells is still under investigation. Hollborn et al. [160] investigated the possible regulation of VEGF in human RPE cells by MMP-9 and MMP-2, showing upregulated expression under hypoxic conditions, as well as an increase in VEGF-A production (but not VEGF-B, VEGF -C, VEGF -D, flt-1, and KDR) and secretion through a direct stimulation of cells by MMP-9, thus facilitating neovascularization. Hoffmann et al. [161] showed that in cultured human RPE, some pro-angiogenic molecules, such as VEGF, could stimulate MMP-2 and MMP-9 secretion from RPE.

It appears that the function of MMPs is not restricted to tissue remodeling, but may also involve the regulation of the complex frame of relations within the microenvironment [162], suggesting that these enzymes are possible therapeutic targets. Thus, some researchers have evaluated their involvement in other pathways related to AMD, such as:

### 4.1. Oxidative Stress

Oxidative stress refers to cellular impairment caused by reactive oxygen species, and is a hallmark of early AMD [163,164]. Seeking to show the possible relationship between oxidative stress and MMPs in AMD, several studies have been carried out with positive results [54,120,148,149,165,166]:Decreased activity of MMP-2 due to the downregulation of MMP-14 and TIMP-2 has been shown after oxidant injury in both in vivo and in vitro studies [120,148,149]. This fact has been linked to the thickening of the BM in the early stages of AMD due to increased deposition of collagen IV.Increased levels of MMP-1 and MMP-3 have been reported after oxidative stress, which could be associated with a shift in the MMP-1,3/TIMP-1 ratio that may provoke increased degradation of collagen I. This has been proposed as one of the mechanisms underpinning CNV development [54].Increased levels of MMP-9 have also been demonstrated after exposure to oxidative stress in ARPE19 cells [165] and human retinal pigment epithelium [166]. In addition, both studies reported an increase in VEGF after oxidative stress [54,166], which could also be associated with wet AMD.

### 4.2. Complement System

Several theories have been proposed regarding the relationship between the complement system and AMD [162,167,168,169,170]. Among them, C3 activation, which is a common endpoint of the three complement pathways (classis, lectin, and alternative), has acquired relevant focus. C3 can be cleaved as much by C3-convertase, resulting in C3b and C3a, as by hydrolysis via tick-over under normal conditions, resulting in C3(H_2_O) [171]. Both C3b and C3(H_2_O) can be deposited in the ECM and are capable of activating the alternative pathway, thereby creating a positive feedback loop. Based on this, Fernandez-Godino et al. [162] studied the growth of normal human fetal RPE cells (hfRPE) on BM obtained from donors with and without AMD. In the donors with AMD, an irregular pattern of collagen IV was reported. The study demonstrated an increase in both MMP-2 activity and levels of C3a in the hfRPE cultivated in the BM of AMD patients. This result could reveal a relationship between abnormal ECM and the complement system.

### 4.3. Renin-Angiotensin System (RAS)

Different studies have been undertaken to elucidate the possible link between hypertension and AMD. Alcazar et al. [43] studied the relationship between RAS and dry AMD, suggesting a possible role for prorenin, which is a precursor of the hormone renin, in the regulation of ECM turnover by increasing the amount of collagen I but without affecting the expression of MMP-2. On the other hand, Striker et al. [172] reported the presence of angiotensin II (ANG II) receptors in EPR, demonstrating a possible link between hypertension and AMD. Using ARPE19 cells, they showed that MMP-2 activity could be induced by increased levels of ANG II, which were also correlated with increased levels of MMP-14 and the degradation of collagen type IV.

### 4.4. High-Temperature Requirement Factor A 1 (HTA-1)

HTA-1 rs11200638 polymorphism has been associated with AMD in multiple genetic studies, although there is still some debate about whether the association is stronger with wet or dry AMD [173,174,175,176]. It is possible that its connection with AMD could be as much by inhibiting the transforming growth factor-b [177], which is seen as an important regulator of extracellular matrix deposition and angiogenesis, as by the digestion of fibronectin-producing fragments, which has been reported to increase the secretion of IL-6, MCP-1, MMP-3, and MMP-9 in murine EPR [178].

Moreover, TIMPs, like MMPs, play an important role in the degradation of ECM components, or deposit accumulation in the BM. Besides the regulatory functions regarding MMPs, TIMP-1 and TIMP-3 have other regulatory properties. In particular, TIMP-1 and TIMP-3 show distinct anti-angiogenic properties by inhibiting microvascular endothelial cell migration [179], or by acting as competitive inhibitors of the binding site between VEGF and VEGF receptor-2 [180]. In particular, high levels of TIMP-3 are associated in AMD with a decreased level of ECM components in the BM [124]. A large number of genome-wide association studies (GWAS) have suggested that the *TIMP-3* gene could be a putative candidate for AMD susceptibility [125]. One of the most relevant clinical examples of TIMP-3 dysfunction and the accumulation of increased levels of TIMP-3 in the BM may be patients with Sorsby’s fundus dystrophy (SFD), a rare autosomal dominant disease with striking similarities to AMD, especially in the late stages, where macular dystrophy, drusen-like deposits, and CNV are primarily seen, and may be misdiagnosed as AMD [181].

TIMP-1, TIMP-3, and TIMP-4 can also take part in the regulation process through the MT1-MMP/TIMP-2 complex [182]. Recently, Krogh-Nielsen et al. [17] reported significant differences in the plasma concentrations of the MMP-9, TIMP-1, and TIMP-3 proteins in AMD patients. They reported higher plasma levels of TIMP-1 and MMP-9 proteins in patients with GA, whereas patients with CNV AMD showed lower plasma levels of TIMP-3, a lower TIMP-3/MMP-2 ratios. Therefore, they hypothesized that an imbalance in the TIMP-3/MMP-2 ratio could be part of CNV pathogenesis.

On the other hand, similarly to MMPs, ADAMs and ADAMTSs are involved in multiple biological processes at the retina level, with important roles in tissue morphogenesis and patho-physiological remodeling, in inflammation and in vascular biology. It has been suggested that these proteins are mainly involved in the inflammatory conditions of the retina characterized by retinal hypoxia and the migration of RPE cells such as AMD, PVR, and DR. In fact, especially in AMD, they are able to compromise the structure of the retinal matrix [135]. In addition, Bevitt et al. [183] observed significant upregulation of these proteins in response to tumor necrosis factor alpha (TNF-α), which is known to play a role in neovascularization. Further studies are needed regarding the potential role of ADAMs and ADAMTSs in pathological neo-angiogenesis.

Analyzing the literature, we have become aware that researchers have had to circumvent several challenges (some of which are stated below), which may sometimes bias the results; this should be borne in mind as we consider the outcomes provided.

Firstly, the issue of obtaining adequate tissue for a study requiring a high level of accuracy, in which several strategies are applied, such as the use of in vitro RPE, animal models, or even human donor eyes, is not trivial, since AMD is a local, progressive, dynamic condition with varying features. The eye animal model is the most widely used, although it does not reproduce all the complexity of AMD [157]. To solve this matter, others studies have proposed the analysis of MMP expression in vivo, obtaining samples from near locations to the retina such as the vitreous or aqueous humour [153,184], as well as the analysis of peripheral samples such as the plasma [53,119], but this approach has sometimes provided inconclusive outcomes.

Secondly, it is important to note that AMD has different stages, in which MMPs could play different roles; not all the studies have taken into account this tenet. Generally, decreased levels of MMP-2 and MMP-9 have been related with the thickening of the BM as well as early stages of AMD, while the overexpression of MMP-2 has been associated with the development of CNV [185].

## 5. Matrix Metalloproteinases and Tissue Inhibitors of Metalloproteinases Gene Polymorphisms Associated with AMD

With advances in sequencing technology in recent decades, the interest in the genetic predisposition to AMD has continued to grow. As a consequence, numerous studies have considered AMD, with over 50% of the heritability explained by two major loci at chromosomes 1q (CFH) and 10q (ARMS2/HTRA1), as the most genetically defined complex disorder [186]. But how this is so has not been completely elucidated, and new approaches with increased attention to rare variants are required. In addition, despite the strong genetic influence on AMD, there is still some controversy over associated versus causative pathological genetic alterations.

Multiple studies have identified several single nucleotide polymorphisms (SNPs) that could modify the risk of AMD [167,168,169]. In general, most studies on genetic markers simply report disease association, and few reflect on severity, disease stages, response to therapy, or disease classification. In addition, the effectiveness of certain biomarkers has often been overestimated, especially in case-control studies, and some reports have yielded conflicting results. However, since therapeutic options are currently limited, AMD-associated SNPs may eventually serve to elucidate the pathways involved in the pathogenesis of AMD and lead to earlier identification and monitoring of high-risk patients. These genetic biomarkers could also serve as powerful tools in designing more informative clinical trials of potential AMD treatments, and in identifying individuals with similar genetic backgrounds. This information may also decrease the cost and discomfort to patients by preventing ineffective and unnecessary treatments and applying potentially high-risk procedures and therapeutics only for those individuals who are most likely to develop advanced AMD.

Within this framework, Fiotti et al. [24] studied the hypothesis that increased repetitions of cytosine-adenine (CA) sequences in the MMP-9 promoter region could enhance its expression and increase the risk of wet AMD, with positive outcomes. As a consequence, several studies were conducted to examine the possibility of different polymorphisms of the MMP genes promoting or decreasing the probability of AMD, with special focus on MMP-2 and MMP-9, as their expression was the most widely studied in previous AMD pathogeneses.

Since then, numerous studies have suggested that polymorphisms of the MMP genes are also associated with an increased risk of AMD [24,25,26,27,28,29,30,31,32,33,34,35,36,37,38]. It has been reported that MMPs are crucial to the regulation of ECM components, and that dysregulation of these pathways could be associated with AMD. Therefore, it is important to study the genetic variants of the MMP genes because they could cause an imbalance in MMP production and lead to an increased risk of AMD [24,25,26,27,28,29,30,31,32,33,34,35,36,37,38].

Table 1 summarizes the polymorphisms of MMP genes that have been analyzed in AMD patients. Polymorphisms in *MMP-1* (rs1799756) [30], *MMP-3* (rs3025058) [26,32], and *MMP-7* (rs11568818) [30,31] genes showed no statistical association with either the risk of AMD or disease progression.

Four polymorphisms of the *MMP-2* gene were analyzed: rs243866, rs243865, rs2287074, and rs2285053 (Table 1) [29,31,32,33,34,35,36,37]. The rs243866 and rs2285053 *MMP-2* genetic variants were not associated with AMD [29,31,32]. Seitzman et al. [37] noted that the A allele of MMP-2 rs2287074 polymorphism was associated with a decreased risk of AMD. The MMP-2 rs2287074 polymorphism is a synonymous variant which has been associated with the risk of breast cancer or osteoporotic bone fracture [187,188]. To confirm the association between MMP-2 rs2287074 polymorphism and AMD, additional studies focusing on larger sets of patients are required. The results of rs243865 MMP-2 polymorphism are controversial. The MMP-2 rs243865 polymorphism consists of a C>T change (-1306C>T) which is located in the *MMP-2* promoter region. It has been reported that this modifies the promoter activity of the *MMP-2* gene [189], and therefore, that it could be related to a reduction in the remodeling and accumulation of basal laminar deposits in AMD patients [86]. Four of the reports did not show any statistical association between rs243865 MMP-2 polymorphism and AMD [33,35,36,37]. Cheng H et al. [29] demonstrated that carriers of the T allele of rs243865 MMP-2 polymorphism were associated with decreased risk of AMD [29]. On the other hand, Liutkeviciene et al. [34] showed that the CC genotype of MMP-2 rs243865 polymorphism was associated with hard drusen in AMD patients [34]. Usategui-Martin et al. [25] performed a meta-analysis to study the association between rs243865 MMP-2 polymorphism and AMD but found none [25]. Further investigations analyzing the combined effect of genetic alterations and environmental factors may improve our current understanding of the association between the rs243865 or other MMP-2 polymorphisms and the risk of AMD, as well as the clinical and biological implications of other risk factors.

Only one genetic variant of the *MMP-20* gene has been studied, rs10895322, whose G allele was found to be associated with the size of eye lesion [38]. The rs10895322 polymorphism is A>G intron variant due to the fact that it can alter the gene expression and therefore modify the neovascular lesion size in neovascular AMD [38]. Finally, four genetic variants of *MMP-9* gene were analyzed (Table 1) [24,27,28,31,32]. Fiotti et al. [24] associated exudative AMD with longer microsatelites in the gene promoter region. The rs142450006, rs3918241, and rs3918242 MMP-9 polymorphisms were associated with an increased risk of AMD [27,28,31,32]. The rs142450006 MMP-9 polymorphism was also associated with progression to choroidal neovascularization [27,28]. It has been reported that MMP-9 could be involved in the degradation of collagen type IV and elastin, resulting in increased levels as a result of aging [137]. Therefore, genetic variants of the *MMP-9* gene could modify its activity and increase the risk of AMD.

It is important to also analyze genetic variants of *TIMP* genes. Table 2 summarizes the polymorphisms of *TIMPs* genes in AMD patients. Two reports evaluated the association between the TIMP-2 rs8179090 polymorphism and AMD [31,36], but only Oszajca et al. [31] found any significant statistical association. The TIMP-2 rs8179090 polymorphism is an upstream variant that has been previously associated with cardiovascular diseases [190,191,192]. The variant could modify the gene expression and therefore modify the risk of the disease, although more in vitro and in vivo studies will be necessary to confirm this hypothesis. To clarify its role as a risk factor for AMD, it will also be necessary to perform studies in larger sets of patients. On the other hand, nine *TIMP-3* gene polymorphisms were studied in AMD patients. The rs6518799, rs756481, rs5749498, rs12170368, and rs1427385 *TIMP-3* genetic variants showed no statistical association with the risk of AMD or disease progression [193,194]. Fritsche et al. [28] reported that the C allele of TIMP-3 rs5754227 polymorphism decreased the risk of AMD. In addition, Kaur et al. [193] reported that the C allele of rs713685 and the G allele of rs743751 *TIMP-3* genetic variants were associated with an increased risk of AMD. The rs5754227, rs713685, and rs743751 polymorphisms are intron variants, and thus, could modify gene expression and increase the risk of the disease. Studies on the TIMP-3 rs9621532 polymorphism, which is also an intron variant, found contradictory results. Three of the reports found no statistical association between the polymorphism and the disease, while four reported that the C allele of TIMP-3 rs9621532 polymorphism was associated with a decreased risk of AMD [125,194,195,196,197,198,199]. TIMP-3 rs9621532 polymorphism, being an intron variant, could be responsible for alteration in the gene expression. To clarify the role of TIMP-3 rs9621532 polymorphism in AMD, it may be necessary to carry out in vitro and in vivo studies to determine the possible relationship between the polymorphism and the pathophysiology of AMD. 

In general, the outcomes of such studies have been inconclusive. We have to bear in mind that these investigations are not without their limitations, the main one being sample size, as already noted by Fritsche et al. [28], who observed that larger samples of patients (over 25,000) than in other complex traits are needed in order to identify rare variants of genes that may have a substantial impact on the risk of AMD. Therefore, we should be wary while analyzing the outcomes in the literature on the association of MMP/TIMP polymorphisms with AMD pathogenesis, seeing that the sample size for a great majority of the studies did not exceed 1000 patients. Thus, further studies are necessary to enhance our understanding of the disease.

In addition, the translation of genetics into biological insights remains a challenge, as the aforementioned carrier MMP polymorphisms do not imply increased MMP expression, free MMPs, or activated forms of the proteins, since numerous processes could interfere in the regulation of MMPs. Consequently, recent studies tried to keep in mind this global view of the pathogenesis, with hypothetical interrelations between different pathways. Budiene et al. [30] investigated the relation of MMP polymorphisms with other gene polymorphisms and with their levels of mRNA and proteins in plasma. Oszajca et al. [31] tried to analyze the possible association between MMP polymorphisms and the expression of some cytokines (IL-1β and IL-6), which have been found to be involved in the pathogenesis of AMD and are a possible crossing point of different pathways. These kinds of approaches, in which the combined effects of genes and ECM environment are taken into account, could enhance our understanding of the disease.

Last but not least, identifying genetic risk factors of *MMP* genes for AMD does not provide better prognoses, since no prophylactic treatment is available for individuals diagnosed either with an imbalance in the MMP brand or with higher risk of AMD. This notwithstanding, we predict that improved knowledge of the genetic predisposition to AMD could be of help to investigators, since it would enhance the selection phase of patients and enable researchers to perform more specific studies, which could consequently provide new and faster outcomes and increase the overall relevance of the research.

## 6. Modulation of Matrix Metalloproteinases Activity in AMD

Although the use of anti-VEGF drugs for the treatment of wet AMD has significantly improved the control of the disease, not all patients benefit from these drugs, especially those with dry AMD, for which there is no efficient treatment. Undoubtedly, identifying additional or alternative therapies that can improve the current standard treatment is very necessary.

Considering the role of MMPs in the progression of AMD, the modulation of their activity could be an interesting therapeutic process. A rational approach to therapy would entail first identifying the matrix components that are responsible for the histopathological changes in the BM and then establishing which MMPs might mediate the effective turnover of these components. In recent years, some therapeutic agents have been used as potential MMP activity modulators in AMD pathology [200,201,202,203,204,205,206,207,208]. However, due to the high degree of structural homology between all members of the MMP family and the complexity of their functions, the results obtained to date with selective MMP inhibitors are not fully satisfactory. In addition, further studies are needed to clarify whether these molecules are safe and effective as monotherapies or as adjuvant treatments in combination with other drugs.

According to data from the literature, there are three main strategies for modulating MMP activity: modulation at the level of transcription, at the level of activation, and at the level of inhibition.

### 6.1. At the Transcription Level

The inhibition of MMPs can be achieved by interfering with the extracellular factors (MMP transcription can be inhibited by corticosteroids or tetracyclines) or by blocking signal transduction pathways (e.g., mitogen-activated protein kinases (MAPK) pathway inhibitors, such as sorafenib and regorafenib, or extracellular signal-regulated protein kinases (ERK) pathway).

Triamcinolone acetonide (TA), a corticosteroid, is one of the first drugs used for the treatment of CNV in AMD patients [209]. TA is able to reduce the expression of MMP-2 and MMP-9, block the migration of choroidal endothelial cells, and inhibit the regulation of feedback between MMP-9 and VEGF in RPE cells under hypoxic conditions [203,204].Doxycycline and minocycline are members of tetracycline antibiotics group; they have strong antimicrobial properties, but also degrees of anti-inflammatory, antiangiogenic, and immunomodulation properties when administered orally or topically [210,211,212]. Doxycycline is the most potent MMP inhibitor among antimicrobial tetracyclines [213,214]; it acts as a noncompetitive inhibitor, interacting with the zinc or calcium atoms in the structural centers of the proteins required for stability [215]. Although the antiangiogenic mechanism of doxycycline is not yet completely understood, it has been reported that doxycycline effectively reduces the progression of CNV by inhibiting the activities of MMP-2 and MMP-9 [210]. Such properties have led to the use of tetracycline as adjuvants in the treatment of AMD. Doxycycline has exhibited promising results by reducing the number of injections required in combination with Anti-VEGF for the treatment of CNV in wet AMD [216], whereas minocycline has been shown to be able to decrease the worsening rate of GA associated with dry AMD [RCT: NCT02564978 and NCT01782989].MAPK inhibitors have been widely studied in the past two decades. However, because of their pharmaceutical limitations and adverse drug reactions, most of these compounds have not moved to clinical trials [217]. Among them, sorafenib and regorafenib seem to be the most attractive MAPK signaling inhibitors of AMD. Both inhibitors have shown antiangiogenic properties targeting multiple pathways such as VEGF receptors 1–3, fibroblast growth factor receptor 1, and platelet-derived growth factor receptor [218,219]. However, even though they have shown effective interference at different levels of the neovascularization cascade and good biocompatibility, the use of sorafenib, which is an oral anti-VEGF, has recently been associated with serious ocular side effects [220]. In the same vein, the RCT: NCT02222207, where regorafenib was used as a topical treatment to inhibit VEGF activity in wet AMD, was terminated after phase IIa, because of its lower efficacy compared to current wet AMD treatments [221].Angiotensin-converting enzyme (ACE) inhibitors and angiotensin receptor blockers (ARBs) have also been proposed for the treatment of AMD due to their pleiotropic effects. Although angiotensin II (Ang II) is able to increase MMP-2 activity and MMP-14 via ERK and p38 in RPE cells, thus inducing changes in the BM which may lead to an increase in subretinal deposits [208], and the use of ARBs can induce regression of choroidal neovascularization in animal models [222,223], recent studies have suggested that these medications do not seem to provide a protective effect against the development of choroidal neovascularization in patients with AMD [224,225].Resveratrol (3,4,5-trihydroxystilbene) was recently studied as a potential therapeutic target for AMD, since it has antioxidant effects against peroxide-induced oxidative stress, reducing MAPK and ERK activation and the expression of cyclooxygenase-2 in RPE cells [226,227,228]. The use of resveratrol for wet AMD has moved to a phase I/II RTC: NCT02625376, where its safety and efficacy in reducing the progression of neovascular AMD were evaluated. Although the completion date was set for 2019, no results have been released so far.

### 6.2. Nuclear Factors of Transcription

Another step towards MMP inhibition could be achieved through their nuclear factors of transcription, such as NF-κB or AP-1 [229]

ω-3 fatty acids: Evidence from animal models and observational studies in humans has suggested that increasing dietary intake of ω-3 fatty acids provides a beneficial effect in the prevention of NF-κB signaling and in the regulation of inflammatory responses in AMD [230]. However, it has been reported that long-standing supplementation in people with AMD does not reduce the risk of progression to advanced AMD or the development of moderate to severe visual loss [230].OT-551 (Othera): This is a disubstituted hydroxylamine with antioxidant properties that operates within the cell to downregulate NF-κB. Despite demonstrating a synergistic effect when used with anti-VEGF treatments in patients with wet AMD [231], the results obtained for the treatment of GA in dry AMD were unimpressive, showing no benefit (RCT: NCT00306488) [232].In this sense, new molecules such as vinpocetine, which inhibits the activation of NF-κB, NLRP3 inflammasome, and cytokine production in RPE cells, could be useful in controlling the chronic inflammation that is believed to drive the degenerative processes in early AMD [233].

### 6.3. MMP Inhibition

The next important step in MMP regulation is their inhibition. Active MMPs can be inhibited by exogenous and endogenous factors, such as nonspecific α-2 macroglobulin, or specifically by TIMP [234]. Anti-MMP antibodies, which can be synthetic or natural inhibitors, are considered as an effective method for MMP inhibition. Several clinical studies have been conducted with these compounds, although several factors, such as low efficacy or the presence of serious side effects, have contributed to disappointing results.

Batimastat (BB-94): This is one of the first drugs subjected to clinical trials. It is a nonselective MMP inhibitor (MMPI) with a broad spectrum of inhibition targets (MMP-1, MMP-2, MMP-3, MMP-7, MMP-9, and ADAM17) which has been shown to suppress neovascularization in animal models at low doses, but at higher doses has been described as toxic in in vivo models [202].Prinomastat (AG-3340): Although a selective MMPI of MMP-2, MMP-3, MMP-9, and MT-MMP1, this drug has only been shown to inhibit angiogenesis in a variety of preclinical models [235,236].BPHA: N-Biphenyl sulfonyl-phenylalanine hydroxamic acid (BPHA) is a selective MMPI that acts on MMP-2, MMP-9, and MMP-14, and has antiangiogenic properties. Oral administration of BPHA has demonstrated a reduction in experimental laser-induced CNV [201].

## 7. Concluding Remarks

Age-related macular degeneration is a multifactorial disorder. Although several molecular pathways are involved, AMD pathophysiology is yet to be fully understood. Recent studies have pointed out that in both early and advanced AMD, the ECM is the area of dynamic changes connected with the activity of its specific regulators, which are composed mainly of MMPs and their tissue inhibitors (TIMPs). The MMP/TIMP complex is crucial for the regulation of the ECM turnover under normal conditions, but under pathological states, its expression can be increased. The dysregulation in the ratio of these factors produces profound changes in the ECM, including thickening and deposit formation, which may eventually lead to AMD development. Additionally, they are a cross point of diverse pathways involved in AMD pathogenesis. In particular, the localization of MMPs in the areas of new vessel formation and in the BM and RPE cells suggests that the MMP/TIMP complex may be cooperatively involved in the early phases of choroidal neovascularization in AMD.

TIMP proteins have also been associated with AMD, mainly in two ways: first, for their central regulatory ability of MMPs, based on which the TIMP/MMP ratio may be a possible marker of the degradation status of the ECM; second, for the anti-angiogenic properties of TIMP-1 and TIMP-3.

The modification of MMP/TIMP expression and activity in human retina may provide clues to the role of the matrix-degrading proteases in the pathogenesis of the complex phenotype of AMD. Indeed, multiple genetic investigations have identified several *MMP* and *TIMP* variants as supposed risk factors of AMD. However, knowledge about MMP and TIMP action in AMD pathogenesis is still controversial, because different studies have demonstrated a protective effect of these enzymes. Another important, unresolved question is if MMPs could be considered as a target of new therapies, either monotherapies or adjuvant treatments. To shed light on this, different approaches have been tried with promising but not fully successful results. Thus, better insight into the pathological mechanisms acting in the area of the ECM may lead to the development of new and improved strategies for AMD treatment.

## Figures and Tables

**Figure 1 ijms-21-05934-f001:**
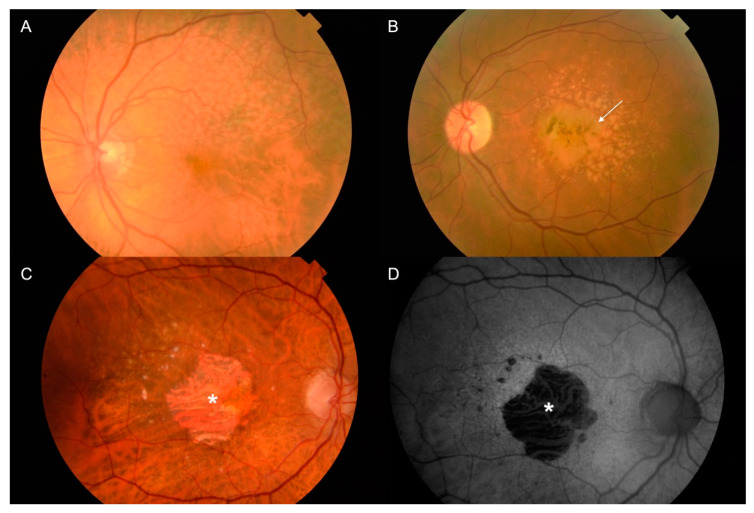
Age-related macular degeneration (AMD) is an eye disease affecting the macula, a central region in the retina. Individuals affected by AMD in its advanced stage may experience a profound loss of central vision. (**A**,**B**) Color pictures of retina with changes typical for early stages of AMD, typified by the presence of numerous large drusen, more or less confluent, and associated (or not) with retinal pigment epithelium (RPE) abnormalities (*arrow*). (**C**,**D**) Color and autofluorescence (AF) pictures of fundus for retina with changes typical for dry AMD. (**C**) The advanced form of dry AMD is typified by the presence of central geographic atrophy (GA) showing a sharply demarcated atrophic lesion of the outer retina, resulting from the loss of photoreceptors, RPE, and choriocapillaris (*asterisk*). (**D**) GA areas typically appear as dark patches in fundus AF images, and can be clearly delineated (*asterisk*).

**Figure 2 ijms-21-05934-f002:**
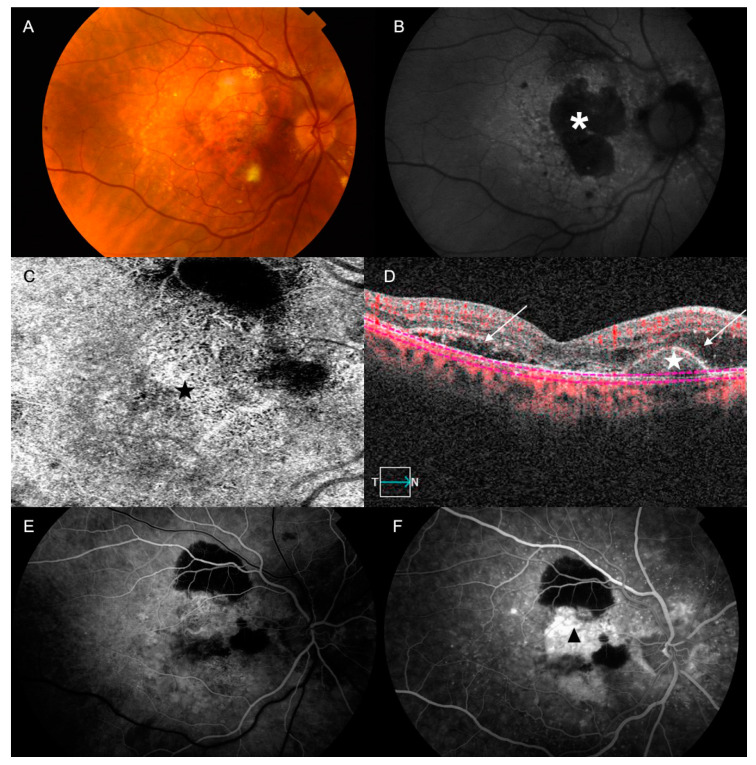
(**A**–**F**) Color, optical coherence tomography (OCT) and fundus fluorescein angiography (FFA) and AF pictures of fundus for retina with changes typical for wet AMD. (**A**) Wet AMD is characterized by abnormal angiogenesis (choroidal neovascularization (CNV)), causing recurrent retinal exudation, subretinal hemorrhage, retinal or pigment detachment and, in the final stages of the disease, subretinal fibrosis (disciform scar). (**B**) AF showing confluent atrophic patches (*asterisk*) with a banded pattern of increased AF in the junction. The CNV can be seen in the OCT-angiography (*black star*); (**C**) the structural OCT enables the identification of the abnormal vascular tree (*white star*) and the presence of subretinal fluid (*arrows*) (**D**). By doing an FFA, we can also confirm the presence of the CNV: (**E**) Early phase: stippled hyperfluorescence with adjacent masking areas by blood or subretinal fibrosis; (**F**) Late phase: The hyperfluorescence increases irregularly due to the presence of progressive leakage (*black arrow-head*).

**Figure 3 ijms-21-05934-f003:**
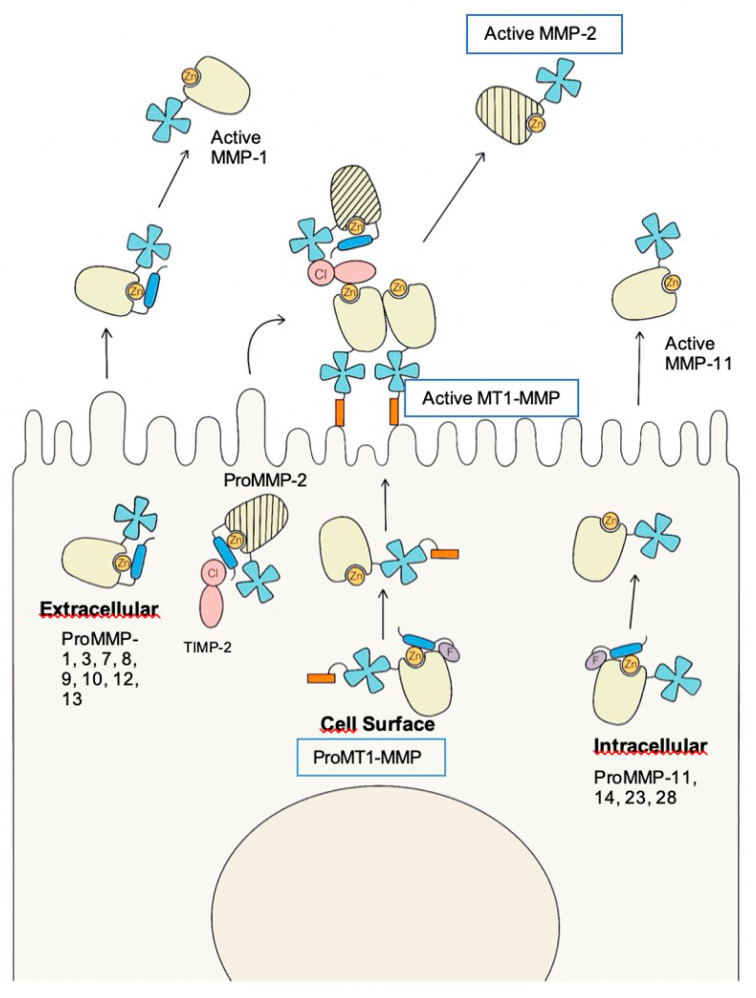
Mechanisms for Pro Matrix Metalloproteinase Activation. ProMMP-2 is the only MMP activated on the cell surface by MT-1MMP (MMP-14); this activation requires the trimolecular complex MT1-MMP/TIMP-2/proMMP-2 and the dimerization of the MT1-MMP. Extracellular activation is applicable to many secreted MMPs, such as proMMP-1,3,7,8,9,10,12, and 13, which are activated by a wide type of proteinases. Furin-activated, secreted proMMPs, such as proMMP-11, 14, 23, and 28 are activated intracellularly due to the removal of propeptides by the action of proprotein convertases such as furin. MMP: matrix metalloproteinases; TIMP: tissue metalloproteinase inhibitor; Cl: C-terminal domain of TIMP-2; F: furin recognition site; Zn: zinc of the active site.

**Table 1 ijms-21-05934-t001:** MMPs polymorphisms associated with AMD.

Gene	Polymorphisms	Authors, Year, Reference	Subjects (*n*)			Association between MMP Genetic Variants and AMD
			AMD Patients		Control Subjects	
			WET AMD	DRY AMD		
*MMP-1*	g.102799766del (rs1799750)	Budiene et al. 2018 [30]	282	–	379	*
*MMP-2*	g.55477894C>T (rs243865)	Cheng, Hao & Zhang 2017 [29]	126		141	CT+TT genotypes were associated with a decreased risk of AMD.
		Seitzman et al. 2008 [37]	802		902	*
		Ortak et al. 2013 [36]	144		172	*
			37	107		
		Liutkeviciene et al. 2016 [35]	387		553	*
		Liutkeviciene et al. 2017 [34]	290		526	CC genotype was associated with hard drusen in AMD patients compared with the control group and soft-drusen group.
			–	34		
		Liutkeviciene et al. 2018 [33]	267	–	318	*
		Usategui-Martín et al. 2019 [25]	1682		2295	*
	g.55493201G>A (rs2287074)	Seitzman et al. 2008 [37]	802		902	The A allele was less prevalent in subjects with AMD.
	g.55477625G>A (rs243866)	Cheng, Hao & Zhang 2017 [29]	126		141	*
	g.55478465C>T (rs2285053)	Liutkeviciene et al. 2015 [32]	148		526	*
		Oszajca et al. 2018 [31]	100	100	100	*
*MMP-3*	g.43784799C>T (rs3025039)	Liutkevičiene et al. 2012 [26]	273		226	*
		Liutkeviciene et al. 2015 [32]	148		526	*
*MMP-7*	g.102530930T>A (rs11568818)	Budiene et al. 2018 [30]	282	–	379	*
		Oszajca et al. 2018 [31]	100	100	100	*
*MMP-9*	CA (13–27) microsatellite	Fiotti et al. 2005 [24]	107		223	Exudative AMD were more frequent in patients with longer microsatellites in the promoter region.
	g.45986354_45986357TTCT (rs142450006)	Fritsche et al. 2016 [28]	16144		17832	The genetic variant was associated with the risk of AMD.
		Yan et al. 2018 [27]	2721			The genetic variant was associated with the progression to choroidal neovascularization.
	g.46007096T>A (rs3918241)	Oszajca et al. 2018 [31]	100	100	100	TT genotype was more frequent in AMD cases, whereas homozygote AA was less frequent.
	g.46007337C>T (rs3918242)	Liutkeviciene et al. 2015 [32]	148		526	CC genotype was more frequent in patients with AMD.
		Oszajca et al. 2018 [31]	100	100	100	CT genotype was more frequent in wet AMD.
*MMP-20*	g.102599525A>G (rs10895322)	Akagi-Kurashige et al. 2015 [38]	–	1146	3248	G allele was associated with increased lesion size

* Statistically association has not been reported.

**Table 2 ijms-21-05934-t002:** TIMPs polymorphisms associated with AMD.

Gene	Polymorphism	Authors, Year, Reference	Subjects (*n*)			Association between TIMP Genetic Variants and AMD
			AMD Patients		Control Subjects	
			WET AMD	DRY AMD		
*TIMP-2*	g.78925807C>G (rs8179090)	Ortak et al. 2013 [36]	144		172	*
			37	107		
		Oszajca et al. 2018 [31]	100	100	100	GC genotype was significantly associated with a protective effect
*TIMP-3*	g.32688525A>C (rs9621532)	Neale et al. 2010 [24]	979		1079	C allele was associated with lower risk of AMD
		Chen et al. 2010 [125]	10049		7148	A allele was associated with increased risk of AMD
		Fauser et al. 2011 [196]	1201		562	*
		Yu et al. 2011 [195]	2594		4134	C allele seemed to have a protective role from the development of AMD
		Zeng et al. 2012 [198]	136	–	181	*
		Ardeljan et al. 2013 [194]	537		921	C allele seemed to have a protective role from the development wet AMD
			189	348		
		Liutkeviciene et al. 2019 [199]	610		306	*
			–	306		
	g.32709831T>C (rs5754227)	Fritsche et al. 2016 [28]	16144		17832	C allele was significantly associated within the control group.
	g.32812451C>T (rs713685)	Kaur, Rathi & Chakrabarti 2010 [193]	250		250	C allele was more frequent in AMD patients.
	g.32838192C>G (rs743751)	Kaur, Rathi & Chakrabarti 2010 [193]	250		250	G allele was more frequent in AMD patients.
	g.32833610G>A (rs6518799)	Kaur, Rathi & Chakrabarti 2010 [193]	250		250	*
	g.32709241A>G (rs756481)	Ardeljan et al. 2013 [194]	537		921	*
			189	348		
	g.32710961G>T (rs5749498)	Ardeljan et al. 2013 [194]	537		921	*
			189	348		
	g.32715261C>T (rs12170368)	Ardeljan et al. 2013 [194]	537		921	*
			189	348		
	g.32848327C>T (rs1427385)	Ardeljan et al. 2013 [194]	537		921	*
			189	348		

* Statistically association has not been reported.

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
