# Peer review of "Matrix Metalloproteinases in Age-Related Macular Degeneration (AMD)"

_ijms, 2020, doi:10.3390/ijms21165934_

Round 1

Reviewer 1 Report

I suggest accepting this manuscript.

Author Response

Many thanks for your input and valuable comments

Reviewer 2 Report

the manuscript is now ready for publication

Author Response

Many thanks for your input and valuable comments

This manuscript is a resubmission of an earlier submission. The following is a list of the peer review reports and author responses from that submission.

Round 1

Reviewer 1 Report

This MS shows excellence scientific value and scientific quality and should be published without any changes.

Reviewer 2 Report

This is a review article regarding an important topic.

Other criteria by which a review can be judged include relevance of material included, critical evaluation and interpretation of that material, syntax, clarity of writing and figures.  

The data compiled in the tables is useful.

However, there is very little rationale for a review on MMP/TIMP and etiology of AMD. 

All the figure panels need arrows to show what is important. Indicate when the panels offer comparisons.

The text must be reviewed by an English speaker. There are endless redundancies, misuse of words throughout the whole manuscript, and paragraphs do not follow in a clearly orderly way. A few of many examples:

Line 16 “

“ pathogenesis of AMD is still fairly 16 understood,” Do the authors mean “barely understood”?

Line 50 “ in a few days’ time”. “A few days “ is sufficient

Line 75 “OCT lets to”. This is unintelligible, as are many other parts of the text

Line 79 “appear to be wide”.. Do the authors mean appear to include?

Lines 563-567 are redundant

Line 568-9 …”beyond question”. This is the only place I see a rationale for the MMP association other than genetic studies.

Reviewer 3 Report

The review is comprehensive and the authors extensively explain the functions and involvement of MMPs and TIMPs. Nevertheless, minor revisions can be performed to make clearer and immediately understandable to the reader the involvement in AMD.

General remarks

Please revise through the manuscript the use of ECM/EMC and MMP/MPP and make sure that the Table numbers are reported correctly.

Figure 1-2

Please indicate the origin of the fundus images: own source or literature

Table 2

This is a very comprehensive table and the amount of information reported is large. Please introduce references in the table corresponding to the information reported (or an extra column with references) to make easier for the reader to retrieve the original literature.

Moreover in the last column (eye implicated processes) it would be nice to have a further division in AMD and non AMD eye processes.

Part 4

While many details are given on the expression and/or activity of the main MMPs and TIMPs found in AMD or RPE cells studies, other details are missing or lacking. For example:

Line 239: thickening due to accumulation of matrix components. Which ones? A general accumulation of components (TIMPs for example) wouldn´t necessarily lead to thickening of the BM.

Line 243: changes in collagens and elastins. Which collagen type? Talking about AMD, it is better to specify which collagen type, considering the switch in collagens types occurs

Line 333: changes in the proteins of extracellular matrix can increase VEGF secretion. Which one exactly are regulating VEGF?

ADAMS and ADAMTS are only shortly mentioned. Knowing the substrates of those enzymes, it would be interesting to place them in AMD context with more details.

Changes in ECM, whether mediated by MMPs or other factors, are an important part of AMD. It would be beneficial to introduce a bit more the connections of those changes with other aspects of AMD directly affected, like complement activation or oxidative stress. Even though, literature on complement is listed (ref 120, 127,128, 129), this knowledge is not explained in the text at all.